# Ancestral Practices for Water and Land Management: Experiences in a Latin American Indigenous Reserve

**David Román-Chaverra, Yolanda Teresa Hernández-Peña and Carlos Alfonso Zafra-Mejía ***

Facultad del Medio Ambiente y Recursos Naturales, Universidad Distrital Francisco José de Caldas, Bogotá E-111711, Colombia; droman@minambiente.gov.co (D.R.-C.); ythernandezp@udistrital.edu.co (Y.T.H.-P.)
* Correspondence: czafra@udistrital.edu.co

**Abstract:** The identification and analysis of mythical images and ancestral practices that make up the ethnos of a community allow us to know its ways of existing in the cosmos. The objective of this paper is to analyze the ancestral experiences associated with the dynamics of socio-environmental management that the *Emberá* Indigenous reserve (Chocó, Colombia) carries out for the conservation of water and land. This study is qualitative and ideographic. We also adopted an ethnographic approach to provide a detailed description of water and land management practices, which correspond to their cultural patterns. Using Atlas Ti V.6.0 software, we identify and analyze these cultural patterns. The results show that the ecosystemic relationships offered by the *Emberá* worldview are part of a true connection with their spiritual world, which fosters respect for the natural elements and understanding of universal natural laws. These relationships are manifested through gifts and penance. The *Emberá* beliefs and religion are a possible methodology for the sustainable management of water and land and, consequently, of the basin where they live. The success of their ethnodevelopment depends significantly on the power figures of their culture: the *Jaibana* (their gods), the elders, and the *Emberá* woman as a cultural agent. The *Emberá* worldview is possibly a valid instrument to enable the sustainable development of modern communities.

**Keywords:** indigenous community; ethnic and cultural diversity; territory; ancestral practice; Latin America



## 1. Introduction

The Chocó region and the Pacific coast of Colombia are recognized in multiple national and international scenarios for their natural and cultural wealth [1–3]. The knowledge, understanding, and strategic valuation of it are key factors that guarantee its protection and conservation. Thus, it is necessary to develop studies that lead us to new alternatives to the Western model of production and consumption that can be integrated with the ancestral knowledge, practices, and beliefs that account for the essential relationships of these Indigenous communities with nature [4]. In fact, we developed this study to visualize new means of environmental management that will allow us to deepen our knowledge and guide future water and land management processes in the communities.

According to Luna [5], from the identification and analysis of mythical images and ancestral practices that make up the ethnos of a community, we learn the essential stories that constitute its own way of existing in the cosmos. These stories reveal an archetypal image of the community's worldview and behavior. Referring specifically to the Indigenous community of study, the interpretation of their ancestral practices and relationship with nature underlies their collective imagination, which is reflected in the symbolic and material complexity of their culture. This territorial connection has implications for the construction of harmonious relationships between human beings and their natural environment [6].

The *Emberá* community is one of the ancestral Indigenous societies that inhabit northern South American and belong to the American linguistic *Chocoes* family, which is related

to the *Arawak*, *Caribbean*, and *Chibcha* families [7]. This study focuses on the *Emberá* that inhabit the *Baudó* and *Bajo San Juan* river basins, which are in the municipalities of *Istmina*, *Alto Baudó*, and *Pizarro* (Chocó, Colombia, South America). This territory is close to the tropical rainforest and mountains of the Colombian Pacific. The study community has a population of 2500 Indigenous people [8].

The main objective of this paper is to perform an analysis of the ancestral experiences associated with the dynamics of socio-environmental management that the *Emberá* community carries out on the *Naucá* River basin (Chocó, Colombia), for the use, management, and conservation of water and land. The results of this study will be relevant for understanding the socioenvironmental processes conducted by this Indigenous community. This study also deepens our knowledge of the sociocultural phenomena that give rise to environmental conservation and the socio-environmental resilience from the beliefs, habits, and ancestral mechanisms that satisfy the needs of the *Emberá* under pressure from acculturation.

## 2. Development of the *Emberá* Community: Change and Resilience Processes

In recent years, geographers such as Cannon and Muller–Mann [9] have argued that the assimilation of the concept of development as growth affects the belief that it can reduce poverty. Furthermore, climate change and its association with development has led to talk of clean development [10], which generates other related debates. For example, does the development-as-growth concept mitigate or contribute to poverty? However, there is evidence that it could possibly increase the exposure of people to risk and exacerbate the vulnerability of poor communities [11]. Thus, the cultural processes that people capabilities depend on for knowledge of their immediate environment are of great importance. The participation of communities is crucial to determining the political and social viability of development since without it a dynamic that projects a sustainable society cannot be established [12]. Max–Neef [13] reports that the application of development models based on theories of economic growth represents a sure route to new frustrations because they ignore the qualitative growth of people and societies. Therefore, it is necessary to rethink the concept of development [14].

In the case of the Chocó region of Colombia, statistics show that the productive activity of the resident communities between 2011 and 2012 was USD 1,783,243 and 1,812,973, respectively. This corresponded to 0.50% of Colombia's GDP (Gross Domestic Product), placing this region among the lowest per capita contributors to the country's economic growth. Ironically, it has some of the world's greatest natural wealth as evidenced by its high levels of biodiversity and water resources [15]. If development is considered with the issue of quality of life, some studies (e.g., [16]) agree that territories with high levels of industrialization have higher levels of pollution and stress that cause the quality of life to deteriorate. Therefore, the level of economic growth of a country is not necessarily an indicator of social improvement.

Economic growth may have overwhelmed the *Emberá* culture and other Indigenous groups in South America, such that economic production and the mass media are leading to cultural homogenization. In this process, Douglas [17] argues that the symbolic world of modern man has less and less to do with his life obligations. Work, like money, is becoming more abstract and less linked to the natural environment. Hence, the socio-environmental dynamic in the *Naucá* River basin (the study community) is possibly mediated by the loss or construction of vital symbolism since the actions of its social actors, like those of any society, are mediated by the symbolic representation of their environment [6].

Additionally, Escobar [18] and Martínez–Alier [19] point out that the traditional knowledge underpinning sustainable survival techniques becomes a productive means to adjust the current capitalist model to develop resilient conditions for ecological balance; namely, the inclusion of strategies such as ecodevelopment, environmental democracy, and environmental governance. In the new global context, the relationship between biodiversity conservation and cultural diversity is more than evident [20]. Leff–Zimmerman [21] reports that technological innovation guided by environmental policies, as well as the productive

management of biodiversity, can come only from the knowledge of culturally rich peoples. However, those who consider the traditional concept of development to be the path to progress, see these countries as of lower rank [22,23].

## 3. Social Resilience under Change Scenarios

According to Adger [24], social resilience is as the ability of a group to respond to external stressors and disturbances from economic, political, and environmental changes, and ecological resilience is the capacity of a natural system to sustain itself in disturbing situations [25]. Adger [24] also points out the close relationship between social and ecological resilience in communities that depend on the natural resources of their immediate environment. This is the case of the *Emberá* community, which has generated synergistic relationships with its environment but is currently undergoing change, stress and disturbance. In relation to globalization induced by development as growth, Rubio [16] describes social fragmentation as one of the processes that inhibits responsible social action, collective decision making and the foundation of linkages between subjects and social subsystems. Furthermore, cultural and moral values disappear when cultural standardization does not allow other avenues for well-being, as is the case with the community under study.

Additionally, Leff–Zimmerman [21] report that social learning is necessary to reconnect with one's own knowledge and the construction of collective meanings and shared identities. In other words, social learning processes are fundamental to ethnodevelopment. Thus, it is necessary for characterizing social dynamics from both productive and cultural perspectives [26]. The International Declaration of San José on Ethnocide and Ethnodevelopment published by UNESCO [27] shows that for Indigenous peoples, land is not only an object of possession and production but the primary basis of their physical and spiritual existence. Moreover, this international declaration indicates that their territorial space is the foundation and reason for their relationship with the universe and the sustenance of their worldview. Based on Batalla and Aravena [28], the worldview of the *Emberá* community with their cumulative history of experience and achievement, are essential to their cultural heritage and are categorical determinants of how their natural environment is managed.

In the mind of the *Emberá* community, the symbolic effects of language, kinship, and social organization condition the cultural values that govern traditional social formation. These values are also unconsciously conditioned by how the community perceives, access and consume their resources [20]. Indeed, the natural environment may be a symbol open to the interpretations of experimental science, but its interpretation also depends on cognitive models based on the cultural knowledge of communities [29,30]. Thus, traditional knowledge must be significantly valued to rediscover how to protect nature. Furthermore, it is possible to consider the expansion and consolidation of the spaces of the *Emberá* culture according to the concept of ethnodevelopment. This can be achieved through the autonomous decision–making to guide the society's own development based on ancestral knowledge [28].

## 4. Methodology

### 4.1. Study Population

We conducted this study on the Indigenous reservations of *Puerto Alegre* and *La Divisa* in the basin of the *Naucá* River, a tributary of the upper basin of the *Baudó* River (Chocó, Colombia; Figure 1). These communities belong to the localities of *La Divisa* and *Naucá* (5°31′33″ N–76°59′44″ W) [3]. *Alto Baudó* was a municipality with a population density of fewer than 20 people/km$^2$ [15]. The *Puerto Alegre* and *La Divisa* reserves had an area of 223,550 hectares and a population of 350. These Indigenous people belong to the *Emberá*, *Baudó*, and *Dóbida* communities, of whom 140 are children and 20 are elderly. The others were adults in a gender–balanced ratio.

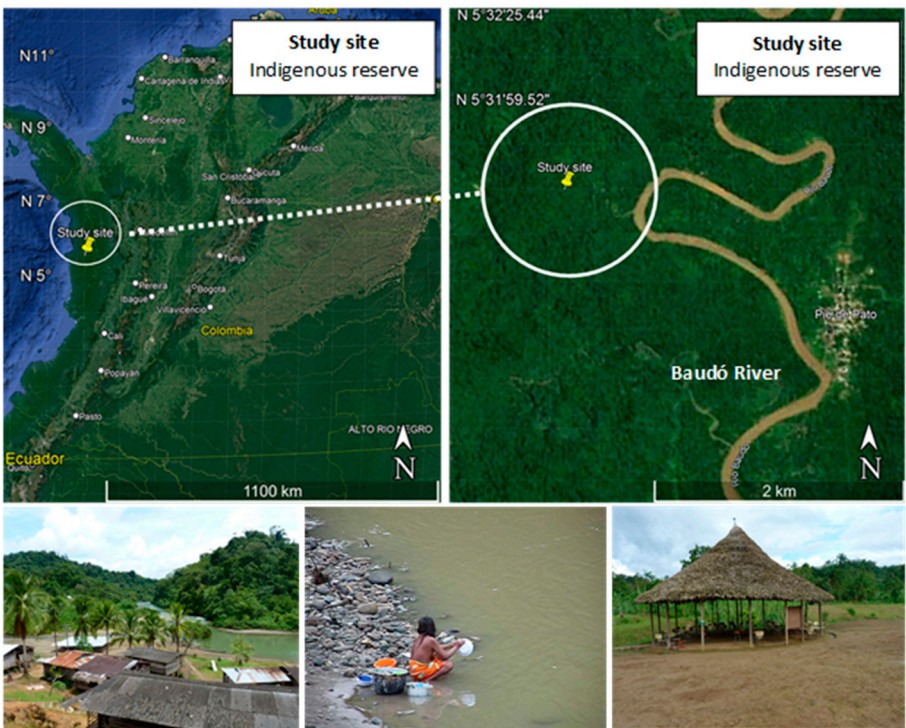

**Figure 1.** Location of the Indigenous community under study.

The area studied was jungle and forest, which was in the region of the equatorial calms and with temperatures above 24 °C (Figure 1). According to Holdridge's life zone classification system [31], this corresponds to a very humid tropical forest (bmh–t) and tropical rainforest (bpt) [3]. According to the IDEAM [32], this region has an average annual runoff between 4000 and 6000 mm because of its high rainfall, the highest in the country and among the highest on the planet. The basin of the *Baudó* River had its source in the *Alto del Buey*, and its main channel had a length of 180 km and an average flow of 200 m$^3$/s (discharged into the Pacific Ocean). The basin is the third-most important in the Chocó region, Colombia (area = 375 km$^2$). The streams that the communities of the Indigenous reserves of *Puerto Alegre* and *La Divisa* influenced were the following: *Naucá*, *Condoto*, *Chirichiri*, *El Rotico*, *El Pedrao*, *Yucal*, and *El Tigre* [3].

*4.2. Information Collection*

This study of the *Emberá* community was qualitative and ideographic [33] because we analyzed the shared ideas that gave meaning to social behavior [34]. An ethnographic approach was also adopted [35] to provide a detailed description of cultural patterns related to water and land management. This approach made it possible to detect community stories related to water and land management and to code them so that they could be analyzed qualitatively [36].

Description, observation, retrospective, and bibliographic reviews were also used as research instruments [37]. The following instruments were used: (i) Deliberate semi–structured interviews from a phenomenological approach [38] and applied to 30 people, mainly elderly and key actors in the *Emberá* community; (ii) a simple survey [39] applied to 13 households on the reserve; (iii) a 5 h video recording [40] that captured different *Emberá* stories; (iv) 325 photographs [41]; (v) a field diary [42] prepared during the study period with observations about community practices and habits; (vi) databases (15 documents) for the collection of secondary information [43] that reported the results of studies related to the *Emberá*. We collected information in the field for 30 days and with the support of an interpreter for interviews with community members.

*4.3. Information Analysis*

The tools used for information analysis were the following: (i) Specialized software Atlas Ti V.6.0 to analyze of social research data [44] and (ii) a matrix of units and categories of analysis [45]. To fulfill of the objectives of this study, the following variables of analysis were defined in relation to water and land management: (i) stories [46] on myths associated with cultural practices; (ii) rituals, offerings, and sacred places [47]; (iii) community habits and customs [48]; (iv) needs met [49] by water and land in the community; (v) factors of change and loss of ancestral knowledge [50].

We established and systematized dialectical codes [51] related to water and land management to identify coincidences and relationships in the ideas expressed by the study community [52]. This systematization made it possible to study the relationships between the established variables to develop cognitive maps [53], which we contrasted with results reported by reference studies related to ancestral knowledge and traditional of natural resource management practices. Moreover, with the dialect codes established, we studied the degree to which a category of analysis (myths, rites, and precepts) related to the total set of associated categories [44].

Table 1 shows the codes generated for the systematic analysis of the dialogues with the inhabitants of the Indigenous reserve in relation to their rites, myths, and precepts for water and land management. Initially, the code depth (grounded) or number of related code chains was analyzed [44]. This made it possible to evaluate the degree of relationship of an analysis variable with the total set of associated variables. Based on the above, we assumed that the codes with the greatest depth were those associated with the management of water and land as fundamental elements for life and satisfying needs, and of sacred importance. The density of relationships or codes with the greatest number of associated relationships was also analyzed [44]. Thus, we assumed that the codes with the highest density were those that referred to the distribution of resources and benefits provided by the land and the river, the location along the river, better territorial and natural resource use, and consideration of water and land as fundamental elements for life. Based on the hypotheses generated from the systematized analysis, we evaluated and discussed the observed relationships among the variables.

**Table 1.** Codes generated for the analysis categories of myths, rites, and precepts in water and land management.

| Code | Grounded | Density |
| --- | --- | --- |
| Water affects the whole community | 4 * | 6 * |
| The water in the reserve is clean | 1 | 2 |
| Water is fundamental for life | 8 * | 5 * |
| Water is sacred | 7 * | 5 * |
| Water is a natural resource | 3 * | 1 |
| Water is a satisfier of needs | 8 * | 4 * |
| Water is used in rituals | 5 * | 3 * |
| Water guarantees food security | 5 * | 2 |
| Water is not exclusive | 2 | 5 * |
| Water comes from rain | 2 | 1 |

**Table 1.** *Cont.*

| Code | Grounded | Density |
|---|---|---|
| Food has become scarce | 3 * | 4 * |
| The *Jaibana*, power figures, and nature | 3 * | 6 * |
| The *Baudó* river is a source of food | 1 | 3 * |
| The river is essential for water to flow | 4 * | 3 * |
| They set up their dwellings along the river adapting them to the floods | 3 * | 4 * |
| The rising of the river can cause floods that affect the fish | 3 * | 1 |
| The distribution of the community along the river allows them a better territorial and natural resource appropriation | 1 | 8 * |
| Drought affects the community | 1 | 3 * |
| Land is fundamental for life | 4 * | 7 * |
| Land is sacred | 8 * | 6 * |
| Land is a resource | 1 | 3 * |
| Land is vital for development | 4 * | 7 * |
| Land guarantees food security | 4 * | 6 * |
| Land should not be usurped | 1 | 7 * |
| Resources from the river are shared with the entire community | 3 * | 8 * |
| The resources and benefits provided by the land are distributed among the members of the community | 1 | 10 * |
| Rivers are a means of communication | 1 | 2 |
| The elders and their ancestral knowledge about water and land are respected | 1 | 3 * |
| Myth of *Ankoré* and the *Atrato* river basin | 4 * | 2 |
| Myth of *Ankoré* and the *Conga Ant* | 2 | 2 |
| Myth of *Ankoré* and the stream | 1 | 2 |
| Myth of the god *Vishiá* | 4 * | 3 * |
| Other species also need water | 1 | 3 * |
| The places with the best conditions for water collection have been identified | 1 | 2 |
| If the forest is preserved, the water will not run out | 2 | 2 |
| Working the land is a reason to celebrate | 2 | 3 * |

Note. * = Most important codes, rating ≥ 3.

## 5. Results and Discussion

### 5.1. Preserving the Basin with Rites and Myths

A well–known *Emberá* myth says that there is an underwater world under the fluvial courses of the *Baudó* River basin. The entrance is through the headwaters, turning the river sources into mystical passageways. The community affirms that these are sacred and protected places. For this reason, access and permanence are restricted. Analyzing the implications derived from this myth suggests that it has induced an environmental management practice that limits activities that may affect the high mountain ecosystems, where the headwaters of the rivers that regulate the water supply in the basin are located.

Another myth narrates that *Vishiá*, a sacred deity associated with the sun, promotes equal access to natural resources and rewards with sunlight the virtuous acts of the community and thus facilitates subsistence work such as planting and hunting (Figure 2). Conversely, if their behavior is contrary to these laws, the community suffers a cold and dreary day of hard labor. The findings suggest that the myth symbolizes cooperation as a group survival strategy. Another *Emberá* myth states that the rivers and streams of the

world originated with the fall of the branches of the *Jenené* tree (water tree). Specifically, the root corresponds to the sea and the branches to the rivers. This myth possibly relates the origin of the *Baudó* River to the origins of the *Emberá* community. According to Salazar [54], the hydrographic network of these basins offered refuge and protection to the *Emberá* during the migrations that occurred due to the Spanish conquest. Therefore, the *Baudó* River and its tributaries represent their place of origin, refuge, and fundamental spaces for life and survival.

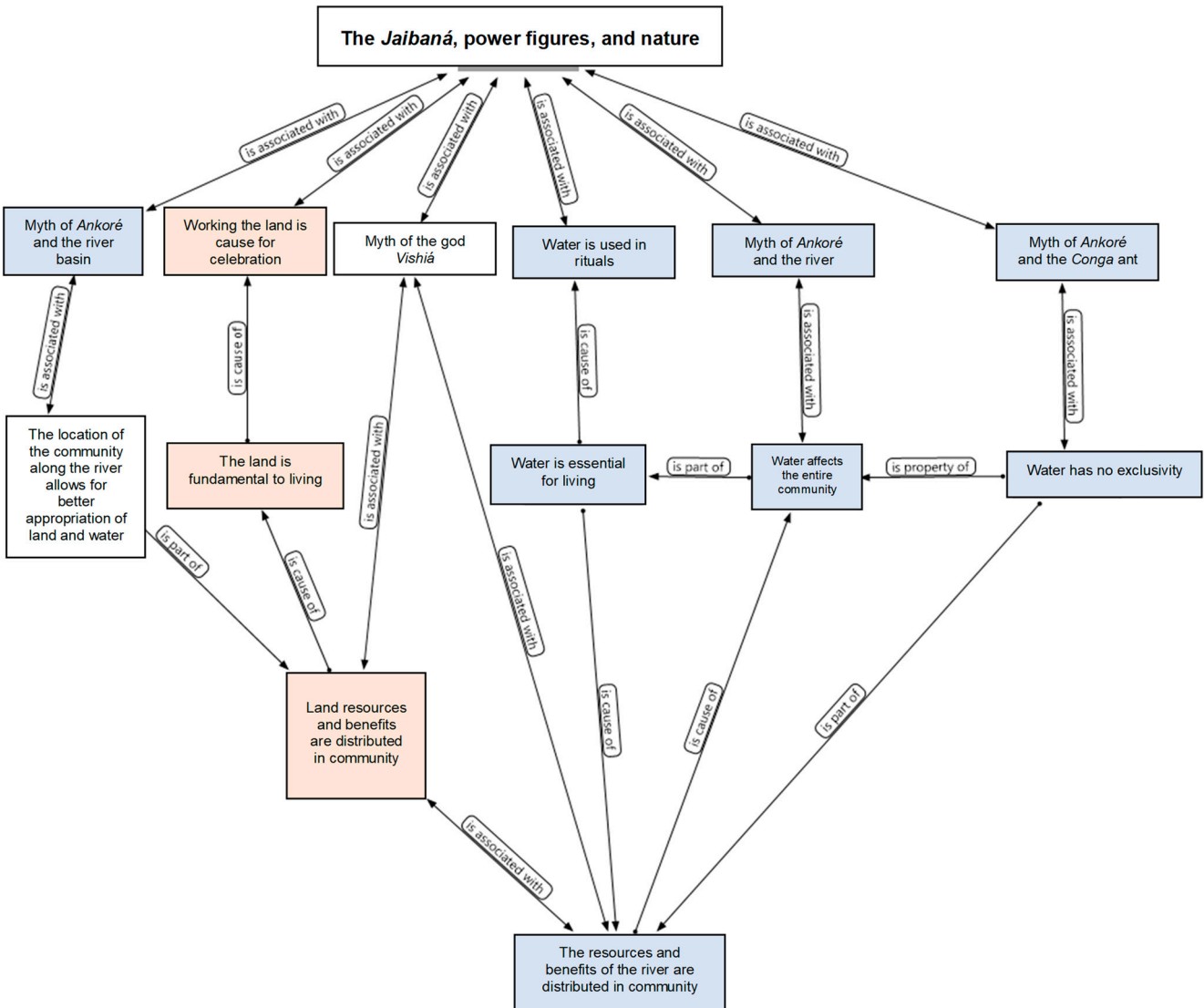

**Figure 2.** Cognitive map of the relationships detected between rites, myths, and traditions for the conservation of the basin in the Indigenous reservation under study: blue = water; orange = land.

Additionally, in the *Jenené* myth, the water was hidden by the *Conga* ant who was punished by the sacred deity *Ankoré*, who compressed her waist to hoard the water for herself (Figure 2). In this story, the animals were also sent to work together to get the water flowing, so the myth promotes the free and equal access to natural resources, the importance of cooperative work as a subsistence strategy, and the fair distribution of benefits. This reasoning is evident in the different subsistence activities, such as cultivating the land, fishing, and hunting. The *Emberá* conduct these complex tasks through group workdays called *Mingas*, which also allow community members to celebrate extraordinary events.

Therefore, the findings hint that the myths evidence a close relationship of the *Emberá* community with water, land, and the other elements of their natural environment. For example, *La agua*, as the *Emberá* call water, is part of their essence as a river community. They insert in their culture multiple symbols associated with this element, the same ones that regulate their daily life, even at a spiritual level. Thus, it is traditional for the community to bathe in the river to symbolize renewal, strength, and spiritual cleansing. Mothers do it during childbirth and girls during their passage to adulthood. Lastly, children from a young age learn to relate to activities associated with water management: swimming, fishing, and sailing. Another *Emberá* myth associated with water describes a river of purification at the point called *Dokarrá* (Root River). They say that through it one embarks on a journey to other worlds after death. For the *Emberá*, there are different cosmic levels linked by a network of waterways that drain from the mountains and reach the Pacific Ocean. According to Hernández [55], the *Bojayá*, *Baudó*, and *Atrato* rivers are considered to be mythical portals to other worlds.

The results suggest that the study community recognizes the rivers as a connecting thread of their territory since through them the *Emberá* communicate upstream and downstream with other communities. In effect, it allows for social and cultural exchange with the communities that inhabit the region. Marzo [56] reports that the rivers also connect with the spiritual world of the gods, which cements their cultural identity, which is understood as the process of using the territory, sharing the same worldview, and feeling part of the same ancestral symbolic construction [8]. Similarly, Barabas [57] found in the Indigenous people of Oaxaca (Mexico) a cultural fabric that physically delimits and spiritually creates components for the symbolic construction of the territory. The *Jaibaná* (the oldest Indigenous man in the community) promotes and safeguards the beliefs associated with cultural patterns that favor the conservation of water sources, proper use of the land, and equitable management of the forest (Figure 2). Thus, the *Jaibaná* acts as a spiritual leader and fulfills the function of regulating the relationship of the *Emberá* community with its natural environment. [58].

The results indicated that the *Jaibaná* also influences the community's daily practices, such as access to hunting areas and water sources (Figure 2). He restricts access using the *Jais* or *Wandras* (jungle spirits) as an element of cultural control in the management of natural resources. In this way, river sources and animal shelters are given special meaning, such as being inhabited by jungle spirits that attack those who dare to desecrate them. They claim that the *Jaibaná* communicates with these spirits and through them induces accidents and diseases to those who do not comply with their laws so that they cannot use the resources and the territory according to particular interests and criteria. On the other hand, the use of resources and territory obeys a logic of collective subsistence that depends on the maintenance and good health of the ecosystems with which they are related. In this study, we based the interpretations described in the previous paragraphs on the analysis of the different relationships detected for the conservation of water and land, and the symbolism expressed in the rites, myths, and traditions of the *Emberá* community.

*5.2. Water and Land Governance*

The *Emberá* perceive water as a satisfactory provider of multiple services such as cooking, food preparation, hydration, making medicines, and treating illness through spiritual rituals. In addition, water is used for making beverages (*Chicha*), fishing, farming, canoeing, personal hygiene, cleaning, celebrations, recreation, and entertainment. Therefore, for the inhabitants of the reserve, water is synonymous with well-being, and the *Baudó* River allows them access to the aforementioned services and provides them with fish for food (Figure 3). This Indigenous community refers to the river as a structural entity of their history and believes that the absence of water would affect them significantly. They also need the water to flow strongly, and for this reason they consider the *Baudó* River to be an entity that conditions their life. This is possibly because the *Baudó* River flows through their entire territory and provides multiple services to this indigenous community. Indeed,

the *Emberá* affirm that their territory must continue to be an essential space for them as well as for other species.

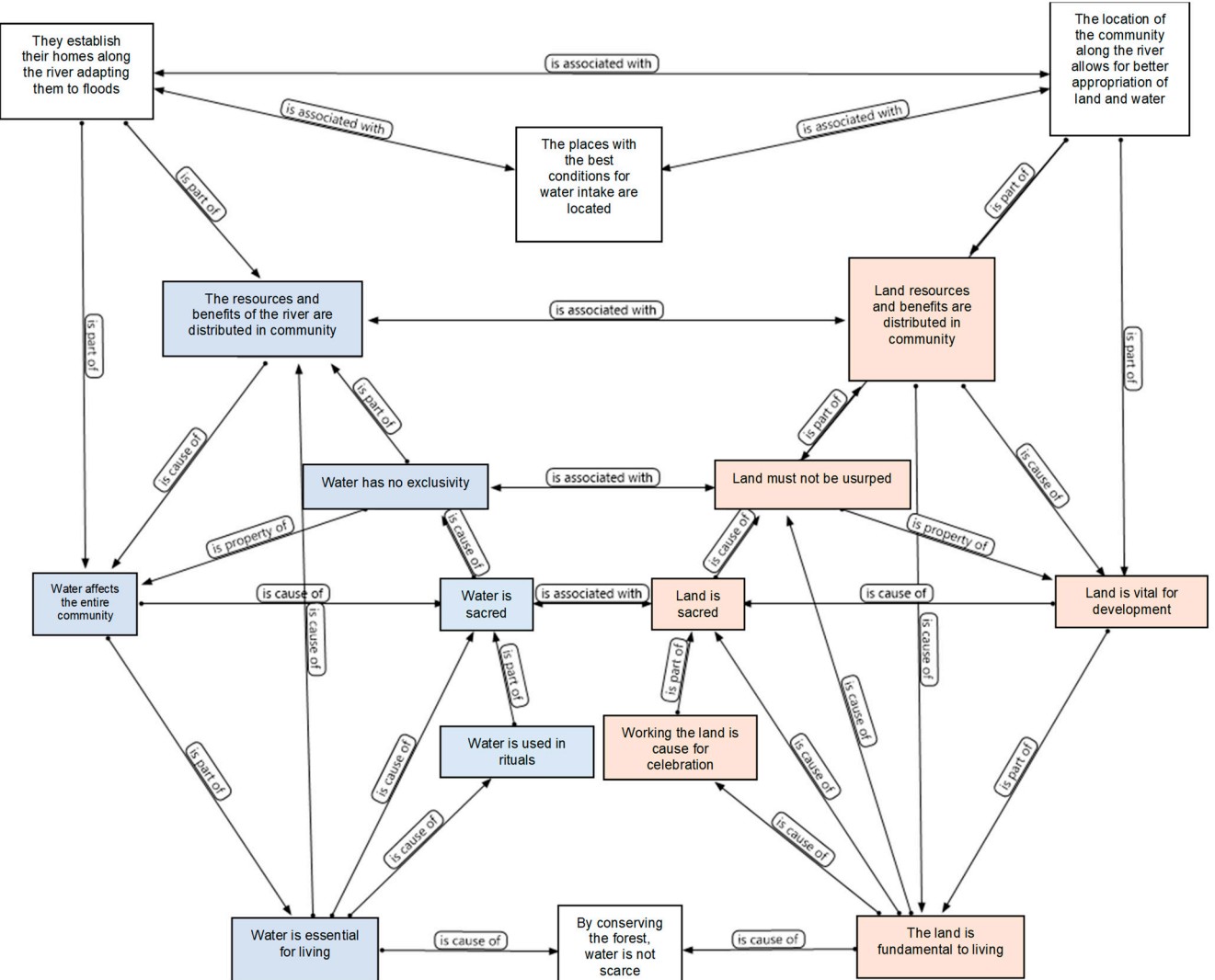

**Figure 3.** Cognitive map on the relationships detected from the precepts associated with water and land management in the Indigenous reservation under study: blue = water; orange = land.

In relation to their perceptions of the land, the *Emberá* affirm that it is useful for finding shelter and security, producing utensils, building houses, cultivating, hunting, eating, working, healing, preparing medicines, and burying the community's dead. Thus, the fertility of the land and the capacity of the forest to provide food and services for subsistence condition the formation of a family and settlement in each place. Through ancestral knowledge, they identify the lands with the greatest potential for a good harvest. To do so, this community analyzes aspects such as soil coloration and the presence of certain plant cover, which are natural indicators for selecting the best places to cultivate. The Indigenous community under study use riparian areas for planting due to their higher productivity and because they claim that the crops do not deteriorate as quickly as those in other cultivation areas. In fact, these areas have the greatest capacity for nutrient recovery due to the hydrosedimentological processes that occur in the river [59].

Additionally, the *Emberá* uses *Socola*, crop rotation, to give the land time to recover (Figure 4). Indeed, this community is aligned with the ecological resilience capacity of the land. The *Emberá* have also identified places that have the best conditions for water collection: small tributaries of the *Naucá* River to which the community restricts access.

They argue that jungle spirits prevent these areas from being deteriorated. Likewise, the *Emberá* are aware of the purity of the untreated rainwater captured from the roofs of their homes through a system of wooden gutters made of riparian plants (*Guadua* spp.). The water is stored in pots and cooking containers, and with this simple technology they benefit from the intense rainfall of the study area (6000–7000 mm per year) [60].

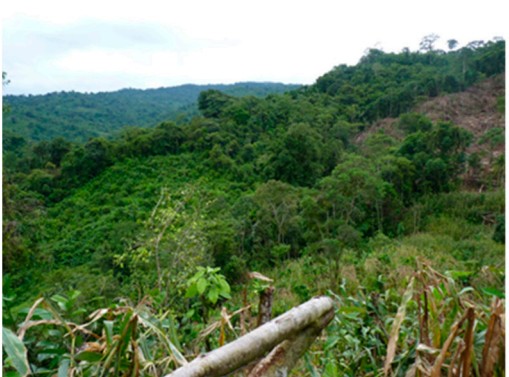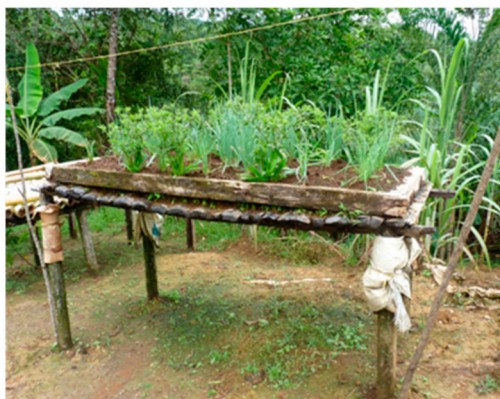

**Figure 4.** Photograph of the crop rotation system (*Socola*) used by the Indigenous community under study.

The results suggest that *Emberá* community members use water equitably. Nevertheless, women have a special role in water management. They are the ones who, among other activities, store and transport water to prepare food and beverages, and wash clothes and kitchen utensils. It is important to highlight another *Emberá* myth: the moon is inhabited by a female deity that randomly and capriciously regulates the occurrence of rains and the passage of water to this world. Thus, we can see that the *Emberá* community considers water and land to be fundamental resources for life and sacred elements since they are indispensable for the realization of their daily tasks (Figure 3).

*5.3. Resistance to Change and Environmental Impacts*

The findings showed that, decades ago, the territorial model of the community under study consisted of the establishment of human settlements along the main streams of the basin. This model was possibly developed because the river and surroundings lands facilitated access to goods and services for the *Emberá* community (Figure 3). Furthermore, it possibly reduced pressure on hunting resources and cultivation areas, as traditionally small *Emberá* family units periodically migrated throughout the forest in search of areas with an adequate food supply. Nevertheless, the custom of settling along the rivers changed due to the establishment of territorial limits for the Indigenous reserve by governmental entities, which implemented a concentric territorial model.

Therefore, the confinement of the Indigenous community to the reserve zone restricted migration. Areas that were once abundant in goods and services developed shortages. There was also a change in the traditional crop rotation cycles, which implied more frequent planting in the same place, which reduced good harvests. Thus, the need for food caused overexploitation of farmland and game species, which lowered the resilience of the reserve's ecosystem. The need for food and the introduction of external cultural stimuli is generating commercial exchanges with occasional visitors from outside the reserve. This new dynamic has been transforming eating and consumption habits and prompted the *Emberá* community to grow crops for profit, further increasing the pressure on their territory. The initiation of commercial relationships with external agents is promoting utilitarian modes of land production and increasing the influence of the outside world. This generates a growing dependency on buying and selling diverse products, which when added to the government's public policies on education, housing, and economic growth, is contributing

to changes in the community's residential and social organization. Sanchez [61] reported a similar trend in other Indigenous communities.

These results suggest an increase in land use conflicts due to the increase in cultivated areas associated with food scarcity. This trend also leads to problems with land use, overexploitation of hydrobiological resources, and water and land pollution by waste and toxic substances, e.g., from the use of outboard motors. Furthermore, the territorial settlement pattern in the form of a nucleated village makes them more vulnerable to flooding, as the population is concentrated on the flood plains. In fact, the Indigenous community under study reports that flooding of the *Naucá* River frequently forces them to relocate their villages.

The findings indicate that colonization also affects the territory of the *Emberá*. Pollution of water resources and regional infrastructure works such as roads fragment biological corridors and cause deforestation, extinction of species and loss of biodiversity, all of which is detrimental to the development of this Indigenous community. Furthermore, the entry of electricity, radio, and television, and the interaction with external communities is transforming water and land management practices and leading to changes in usage habits of the watershed ecosystem. We also observed the introduction of the concepts of private property and poverty, associated with the desire to abandon the reserve and its ancestral customs. The influence of outside education on children is another factor that is changing their lifestyle and management of the natural environment. The introduction of new education models can lead to the loss of cultural identity and the forgetting of ancestral knowledge. In this sense, *Emberá* elders argue that Indigenous people had more value for what they knew because children are being educated to help their parents economically from outside the reserve. Lastly, drug trafficking and illegal mining brings conflict over territorial control between illegal armed groups and the military, producing insecurity in the community about its future and well-being.

Additionally, evangelization has contributed to the loss of values and transformed the community's way of life. Morales [62] reports that the loss of traditional knowledge and culture is related to the influence of the government and the Christian church. Governmental influence, however, is necessary for *Emberá* territory to achieve social recognized, so the government requires the *Emberá* to participate in elections. This governmental model forces this and other communities to adopt lifestyles that contribute to the generation of income but diminish cultural values related to cooperation and the equitable distribution of the watershed's natural resources. This trend also generates conflicts of interest over land ownership and management. In relation to the cultural expressions of the *Emberá*, it is pertinent to recall what Lindón said [63]: behind the ways of saying and doing that we have learned, there is in every culture a way of approaching the world, treating nature, living in the present, and projecting the future. Therefore, this ancestral knowledge in Indigenous communities must be recognized and respected by all governments.

The knowledge of the natural environment expressed through the culture of the *Emberá* is evidence that there is an understanding the world that has a sustainable vision of the ecosystem of the basin. In other words, they have a simple lifestyle, not because it is an easy way to survive but because any other means of using the environment to facilitate life makes harmonious relations with the natural environment unsustainable [13]. This does not mean to that this Indigenous community will be removed from Western knowledge; rather, modern knowledge must respect its cultural identity and for this it must be nourished by it. Lastly, de Sousa Santos [64] reports that it is necessary to bring out an ecology of knowledge: the interaction between modern scientific knowledge and the non–Western knowledge of indigenous communities.

## 6. Conclusions

The findings of this study suggest that an understanding of ecosystemic relationships offered by the *Emberá* cosmovision probably comes from a true connection with their spiritual world. This worldview fosters respect for natural elements and an understanding

of universal natural laws, which is evident through gifts and penances. In other words, the beliefs and religion of the *Emberá* are a possible methodology for the sustainable management of water and land, and consequently of the basin ecosystem.

Additionally, the results indicated that an understanding of the relationship between man and nature has allowed the *Emberá* community to manage the resources of the tropical forest of the *Baudó* mountain range. This sustainable management of water and land is evident in their mythology, rituals, and cultural precepts, a social amalgam that manages to endure. Therefore, it is relevant to recognize that a community's sense of attachment to the water and land is extremely important for managing their environment. Thus, we must highlight cultural visions that recognize the significance of the relationship between nature and the spiritual world. Moreover, it is necessary to find ways to ensure that modern development and progress do not homogenize or weaken these Indigenous communities. Indeed, this leads us to create spaces for outreach to maintain these communities and their ancestral cultural identities.

The findings imply that the success of *Emberá* ethnodevelopment depends significantly on the power figures of the culture: the *Jaibana* (their gods), the elders, the *Emberá* woman as a cultural agent, and their ethno-ecology. This success demonstrates the ability to adapt to modern crises and generate strategies of social organization and consumption that satisfy needs while promoting environmental sustainability. Thus, when the intangible effect of that which is contained in ancestral knowledge and which sustains values consistent with the equitable, responsible, and sustainable management of natural resources is recognized and protected, a management tool is put in place through culture that can be a global reference for other communities. Therefore, these findings show that a more harmonious relationship between man and nature must recognize scientific knowledge as a valid instrument to make possible the sustainable development of modern communities.

**Author Contributions:** Conceptualization, D.R.-C. and Y.T.H.-P.; Methodology, D.R.-C., Y.T.H.-P. and C.A.Z.-M.; Software, D.R.-C.; Validation, D.R.-C., Y.T.H.-P. and C.A.Z.-M.; Formal analysis, D.R.-C. and C.A.Z.-M.; Investigation, D.R.-C. and Y.T.H.-P.; Resources, D.R.-C., Y.T.H.-P. and C.A.Z.-M.; Data curation, D.R.-C., Y.T.H.-P. and C.A.Z.-M.; Writing—original draft, D.R.-C. and Y.T.H.-P.; Writing—review & editing, C.A.Z.-M.; Visualization, D.R.-C. and C.A.Z.-M.; Supervision, Y.T.H.-P.; Project administration, D.R.-C. and Y.T.H.-P.; Funding acquisition, D.R.-C. and Y.T.H.-P. All authors have read and agreed to the published version of the manuscript.

**Funding:** This research received no external funding.

**Informed Consent Statement:** Informed consent was obtained from all subjects involved in the study.

**Acknowledgments:** The authors thank the *Emberá* community of Puerto Alegre (Chocó, Colombia), John Von Neumann Pacific Environmental Research Institute—IIAP (Colombia), and INDESOS and GIIAUD Research Groups of the Universidad Distrital Francisco José de Caldas (Colombia).

**Conflicts of Interest:** The authors declare no conflict of interest.

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
