# Peer review of "Ancestral Practices for Water and Land Management: Experiences in a Latin American Indigenous Reserve"

_sustainability, doi:10.3390/su151310346_

Round 1

Reviewer 1 Report

The authors report on an ethnographic study of a small indigenous group that lives within the confines of a reserve on its traditional lands in the Colombian rain forest. The study aimed at describing the group’s relationship with its environment as mediated by social, cultural, and spiritual knowledge and beliefs, specifically to identify linkages between the community and the sustainability of their land and water resources. The results are purported to support consideration of traditional knowledge in development and land/water management. The paper contains many interesting and intriguing observations; however, the presentation fails to make a coherent case. Evidence is not clearly presented, and analysis does not lead clearly to conclusions. The writing is dense, unnecessarily complicated, and often ungrammatical. The figures should indicate whose cognitive maps they illustrate. (Do they map how the Emberá themselves think of their land and water resources?) The results should be presented without being intermixed with conclusions, the derivation of the cognitive maps should be explained, the analytical links between evidence and conclusions should be spelled out. The various roles of change agents should be stated without recourse to an uninformative passive voice. It is disappointing that the primary work does not stand out because of the confused style in which it is presented.

See above.

Author Response

Reviewer 1

We appreciate all the comments made by the referees. These made it possible to improve the paper. Below you will find the answers to each of your comments.

Comments and Suggestions for Authors:

The authors report on an ethnographic study of a small indigenous group that lives within the confines of a reserve on its traditional lands in the Colombian rain forest. The study aimed at describing the group’s relationship with its environment as mediated by social, cultural, and spiritual knowledge and beliefs, specifically to identify linkages between the community and the sustainability of their land and water resources. The results are purported to support consideration of traditional knowledge in development and land/water management. The paper contains many interesting and intriguing observations; however, the presentation fails to make a coherent case. Evidence is not clearly presented, and analysis does not lead clearly to conclusions. The writing is dense, unnecessarily complicated, and often ungrammatical. The figures should indicate whose cognitive maps they illustrate. (Do they map how the Emberá themselves think of their land and water resources?) The results should be presented without being intermixed with conclusions, the derivation of the cognitive maps should be explained, the analytical links between evidence and conclusions should be spelled out. The various roles of change agents should be stated without recourse to an uninformative passive voice. It is disappointing that the primary work does not stand out because of the confused style in which it is presented.

  1. The writing is dense, unnecessarily complicated, and often ungrammatical. 

Response: We accept the referee's suggestion. Therefore, all the wording of the manuscript was revised and adjusted. All changes made were indicated with the Word tracking control or left in red text.

  1. The figures should indicate whose cognitive maps they illustrate. (Do they map how the Emberá themselves think of their land and water resources?)

Response: We accept the referee's suggestions. We adjusted the titles of figures 2 and 3. Additionally, we added table 1 to the article. This is to support the relationships detected and shown in these figures.

  1. The results should be presented without being intermixed with conclusions.

Response: We accept the referee's suggestions. Thus, the wording of the results and discussion chapter was adjusted. In effect, the conclusions of the article were adjusted.

  1. The derivation of the cognitive maps should be explained.

Response: We accept the referee's suggestion. For this purpose, we include Table 1. Please check L227-241.

  1. The analytical links between evidence and conclusions should be spelled out.

Response: We accept the referee's suggestion. Thus, we added Table 1 to support the relationships shown in Figures 2 and 3 (results and discussion). Additionally, all conclusions were adjusted.

  1. The various roles of change agents should be stated without recourse to an uninformative passive voice.

Response: We accept the referee's suggestion. Thus, the wording of the chapter on results and discussion was adjusted.

Reviewer 2

We appreciate all the comments made by the referees. These made it possible to improve the paper. Below you will find the answers to each of your comments.

Comments and Suggestions for Authors:

This work presents is an interesting topic. However, I am not sure about the impact of their result for the scientific community based on the scope of sustainability journal.

  1. The methodology section seems to be well-based, nevertheless, it is not clear enough how the authors did all the analysis and how they got their results. Besides, the results are not strong enough and I wonder how those results are useful for the scientific community, and not only for the study area.

Response: We accept the referee's suggestions. For this, we add in detail how the data were analyzed. This is included in the chapter on materials and methods. Please see L227-241 and Table 1. Additionally, we proceeded to improve the discussion of results and adjusted the conclusions to visualize the practical usefulness of this study in a global context. All changes made are marked with Word tracking control or red-colored text.    

  1. Figure 1 must be improved, it is poor quality and the study population section is not well described, and the explanation on page 4 is not clear.

Response: We accept the reviewer's comments. We proceeded to improve Figure 1 and adjusted section 2.1. of the manuscript.

  1. Figure 2 and 3, how did they get to those relationships? 

Response: We accept the referee's comment. The procedure used to detect the relationships shown in Figures 2 and 3 was included in the materials and methods chapter. Please review L227-241 and Table 1.

  1. How did each author contribute to this work? There is a lot of missing information about the entire paper and its results.

Response: We accept the reviewer's request. We proceeded to include a section in the article on the contribution of each author. L505-511.

Reviewer 3

We appreciate all the comments made by the referees. These made it possible to improve the paper. Below you will find the answers to each of your comments.

Comments and Suggestions for Authors

The manuscript deals with a highly topical and relevant subject, it is structured in a principally comprehensible way (more on this below) and linguistically comprehensible. The core ideas of the text become clear. Figures 2 and 3 are very instructive. There are some aspects that I think could be improved:

  1. There is frequent mention of "identity" or "cultural identity." Since both terms have a large semantic yard, it should be made clear what the authors mean by them.

Response: We consider the referee's suggestions. Please see L299-301.

  1. Figure 1 includes a satellite image where it is hardly clear what it is meant to represent and why. Here it is strongly recommended to create a separate map showing those things that may also be of interest to readers of the paper. Especially those that are not local.

Response: We accepted the referee's suggestions. Please see Figure 1.

  1. It should be made clearer on what theoretical basis the text stands. Although "construction" is often mentioned, the text should nevertheless be linked to a theoretical discourse, also in order to generate more coverage.

Response: We accepted the referee's suggestions. Please see L46-51, L101-105, L134-136, and L299-301.

  1. The introduction is very extensive. And sections 1.1and 1.2 are no longer really introductory. Here, consideration should be given to not separating these out into a separate chapter.

Response: We consider the arbitrator's comment. However, we did not clearly understand his objective. We possibly decided not to separate these sections to maintain a similar version of the manuscript for the four assigned referees.

  1. The conclusion should be revised again to present to an international audience what the key messages are from the study. This would also facilitate the dissemination of the article's contents.

With these adjustments, the article should also receive the resonance it deserves as a result of its qualities. Thus, it is more a matter of increasing connectivity than of fundamentally changing anything.

Response: We accept the referee's comment. Thus, all conclusions were adjusted.

Reviewer 4

We appreciate all the comments made by the referees. These made it possible to improve the paper. Below you will find the answers to each of your comments.

Comments and Suggestions for Authors

ANCESTRAL PRACTICES FOR WATER AND LAND MAN-AGEMENT: EXPERIENCES IN A LATIN AMERICAN INDIG-ENOUS RESERVE

This is an important paper in a number of ways.  Firstly, because we have been seeking to manage water for millennia and in many cases very successfully.  We need to learn how this was done and not, as is too often the case, to assume that we necessarily know better than our inherently ‘primitive’ ancestors.  Secondly, because as a piece of anthropological research, it focuses on the way a culture interprets the world and consequently defines the way in which we seek to manage it.  Using the anthropological definitions of culture, such as that of Geertz, each discipline is a culture.  Thus, traditionally, engineers defined their role as to determine what the public needed, to determine what was the best means of modifying the environment to meet those needs, and to implement that means.  Similarly, economists have defined the world in terms of competition and markets, failing, as North noted, to provide either an economic theory of competition or to define when cooperation would be more efficient than competition.  The paper describes a culture of co-action and mutual support.

As a piece of research centred on water management, I have a few questions directed from both directions.  As a study using the techniques of anthropology, there are a couple of questions that follow:

  1. How long was your stay in the area?

Response: We consider the referee's suggestions. Please see L201-203.

  1. In what language(s) were the interviews conducted?

Response: We accepted the referee's suggestions. Please see L201-203.

  1. From the perspective of water management, an expansion of text on water availability would be helpful to the reader.  From what I found, intra-year variability in rainfall intensity is low (https://weather-and-climate.com/average-monthly-precipitation-Rainfall,nuqui-choco-co,Colombia) but I would suspect that inter-year variability is higher in consequence of the El Nino/La Nina cycle http://www.ideam.gov.co/web/siac/ninoynina.  Once average annual rainfall surpasses arid conditions, it’s the variability that has to be managed.  At l235, the text refers to a well-known Emberá myth that there is an underwater world under the surface water.  I wondered whether, given what is know now about the underground flows well beneath the river Amazon, whether this might be supported by geological conditions (https://www2.sgc.gov.co/LibroGeologiaColombia/tgc/sgcpubesp37201914.pdf).  Does rainfall result in any groundwater recharge?  The paper is thin on the actual practices they adopt to cope with the extreme rainfall and possible variability in water availability.

Response: We accepted the referee's suggestions. Please see L173-175, L248-255, L360-368, and figures 2 and 3.

  1. The paper contains an impressive list of references but needs those parts of the guidelines for authors which have been included to be addressed before being edited out.

Response: We accepted the referee's suggestions. Please see the references chapter.

  1. l56 typo, I think.  The figure I found was that the indigenous population of Colombia is made up of 87 ethnic groups totalling 1.45 million people  (https://www.colombia.co/en/colombia-country/colombias-indigenous-groups/#:~:text=With%20some%2087%20ethnic%20groups,3.5%25%20of%20the%20total%20population.)

Response: We consider the referee's suggestions. However, this study is exclusively about the Emberá indigenous population, and not about the entire indigenous population of Colombia.

  1. l408 flood rings? What are these?

Response: We accepted the referee's suggestions. However, please see https://www.frontiersin.org/articles/10.3389/fpls.2016.00775/full

  1. l428 armed conflict.  This remark is rather understated.  When I reached 40, and planning another visit to Colombia, I was pleased to have achieved 40 as the leading cause of death amongst males aged 18-40 was then violence.  What are the chief threats in the Emberá area? 

Response: We accepted the referee's suggestions. Please see L440-442.

  1. Very good apart from 'flood ring' which probably a direct spanish-engish translation.

Response: We accepted the referee's suggestions. However, please see https://www.frontiersin.org/articles/10.3389/fpls.2016.00775/full

Reviewer 2 Report

This work presents is an interesting topic. However, I am not sure about the impact of their result for the scientific community based on the scope of sustainability journal.

The methodology section seems to be well-based, nevertheless, it is not clear enough how the authors did all the analysis and how they got their results. Besides, the results are not strong enough and I wonder how those results are useful for the scientific community, and not only for the study area.

Figure 1 must be improved, it is poor quality and the study population section is not well described, and the explanation on page 4 is not clear.

Figure 2 and 3, how did they get to those relationships? 

How did each author contribute to this work? There is a lot of missing information about the entire paper and its results.

Author Response

(The authors gave the same response as above.)

Reviewer 3 Report

The manuscript deals with a highly topical and relevant subject, it is structured in a principally comprehensible way (more on this below) and linguistically comprehensible. The core ideas of the text become clear. Figures 2 and 3 are very instructive.
There are some aspects that I think could be improved:
1) There is frequent mention of "identity" or "cultural identity." Since both terms have a large semantic yard, it should be made clear what the authors mean by them.
2) Figure 1 includes a satellite image where it is hardly clear what it is meant to represent and why. Here it is strongly recommended to create a separate map showing those things that may also be of interest to readers of the paper. Especially those that are not local.
3) It should be made clearer on what theoretical basis the text stands. Although "construction" is often mentioned, the text should nevertheless be linked to a theoretical discourse, also in order to generate more coverage.
4) The introduction is very extensive. And sections 1.1and 1.2 are no longer really introductory. Here, consideration should be given to not separating these out into a separate chapter.
5) The conclusion should be revised again to present to an international audience what the key messages are from the study. This would also facilitate the dissemination of the article's contents.
With these adjustments, the article should also receive the resonance it deserves as a result of its qualities. Thus, it is more a matter of increasing connectivity than of fundamentally changing anything.

At one point or another, the choice of words is similar, which does not detract from the quality of the content, but it does make for somewhat tedious reading.

Author Response

(The authors gave the same response as above.)

Reviewer 4 Report

 ANCESTRAL PRACTICES FOR WATER AND LAND MAN-2 AGEMENT: EXPERIENCES IN A LATIN AMERICAN INDIG-3 ENOUS RESERVE

This is an important paper in a number of ways.  Firstly, because we have been seeking to manage water for millennia and in many cases very successfully.  We need to learn how this was done and not, as is too often the case, to assume that we necessarily know better than our inherently ‘primitive’ ancestors.  Secondly, because as a piece of anthropological research, it focuses on the way a culture interprets the world and consequently defines the way in which we seek to manage it.  Using the anthropological definitions of culture, such as that of Geertz, each discipline is a culture.  Thus, traditionally, engineers defined their role as to determine what the public needed, to determine what was the best means of modifying the environment to meet those needs, and to implement that means.  Similarly, economists have defined the world in terms of competition and markets, failing, as North noted, to provide either an economic theory of competition or to define when cooperation would be more efficient than competition.  The paper describes a culture of co-action and mutual support.

As a piece of research centred on water management, I have a few questions directed from both directions.  As a study using the techniques of anthropology, there are a couple of questions that follow:

·       How long was your stay in the area?

·       In what language(s) were the interviews conducted?

From the perspective of water management, an expansion of text on water availability would be helpful to the reader.  From what I found, intra-year variability in rainfall intensity is low (https://weather-and-climate.com/average-monthly-precipitation-Rainfall,nuqui-choco-co,Colombia) but I would suspect that inter-year variability is higher in consequence of the El Nino/La Nina cycle http://www.ideam.gov.co/web/siac/ninoynina.  Once average annual rainfall surpasses arid conditions, it’s the variability that has to be managed.  At l235, the text refers to a well-known Emberá myth that there is an underwater world under the surface water.  I wondered whether, given what is know now about the underground flows well beneath the river Amazon, whether this might be supported by geological conditions (https://www2.sgc.gov.co/LibroGeologiaColombia/tgc/sgcpubesp37201914.pdf).  Does rainfall result in any groundwater recharge?  The paper is thin on the actual practices they adopt to cope with the extreme rainfall and possible variability in water availability.

The paper contains an impressive list of references but needs those parts of the guidelines for authors which have been included to be addressed before being edited out.

Odd queries:

l56 typo, I think.  The figure I found was that the indigenous population of Colombia is made up of 87 ethnic groups totalling 1.45 million people  (https://www.colombia.co/en/colombia-country/colombias-indigenous-groups/#:~:text=With%20some%2087%20ethnic%20groups,3.5%25%20of%20the%20total%20population.)

l408 flood rings? What are these?

l428 armed conflict.  This remark is rather understated.  When I reached 40, and planning another visit to Colombia, I was pleased to have achieved 40 as the leading cause of death amongst males aged 18-40 was then violence.  What are the chief threats in the Emberá area? 

Conclusion:

Publish with some edits and clarifications.

very good apart from 'flood ring' which probably a direct spanish-engish translation.

Author Response

(The authors gave the same response as above.)

Round 2

Reviewer 1 Report

New description of analytical methods and new conclusions make the paper acceptable. 

The style is still complicated and dense, which detracts from the overall impact, especially on readers from other disciplines; however, readers steeped in the cited literature may not find the style a problem. Because of the complex sentence structure, there are still occasional grammatical errors.

Author Response

Comments:

  1. The style is still complicated and dense, which detracts from the overall impact, especially on readers from other disciplines; however, readers steeped in the cited literature may not find the style a problem. Because of the complex sentence structure, there are still occasional grammatical errors.

Response: We double-checked the entire wording of the article. The changes are marked with the Word tracking control.

Reviewer 3 Report

From my point of view, the notes have been adequately taken into account. I would continue to keep sections 1.2 and 1.3 as a separate chapter, as they are more than introductory, but this does not detract from the quality of the paper.

Author Response

Comments:

  1. From my point of view, the notes have been adequately taken into account. I would continue to keep sections 1.2 and 1.3 as a separate chapter, as they are more than introductory, but this does not detract from the quality of the paper.

Response: We adjusted the article based on the referee's comment.